# Identification and characterization of a novel multi-stress responsive gene in Arabidopsis

**Faiza Tawab[1], Iqbal Munir[1]\*, Zeeshan Nasim[1], Mohammad Sayyar Khan[2], Saleha Tawab[3], Adnan Nasim[3], Aqib Iqbal[1], Mian Afaq Ahmad[1], Waqar Ali[4], Raheel Munir[1], Maria Munir[1], Noreen Asim[2]**

**1** Division of Biochemistry, Institute of Biotechnology and Genetic Engineering (IBGE), The University of Agriculture, Peshawar, Khyber Pakhtunkhwa, Pakistan, **2** Genomics and Bioinformatics Division, Institute of Biotechnology and Genetic Engineering (IBGE), The University of Agriculture, Peshawar, Khyber Pakhtunkhwa, Pakistan, **3** Agriculture Research System, Peshawar, Khyber Pakhtunkhwa, Pakistan, **4** Department of Biotechnology, University of Malakand, Chakdara, Lower Dir, Khyber Pakhtunkhwa, Pakistan

\* iqmunir@aup.edu.pk

## Abstract

Abiotic stresses especially salinity, drought and high temperature result in considerable reduction of crop productivity. In this study, we identified *AT4G18280* annotated as a glycine-rich cell wall protein-like (hereafter refer to as *GRPL1*) protein as a potential multistress-responsive gene. Analysis of public transcriptome data and GUS assay of *pGRPL1::GUS* showed a strong induction of *GRPL1* under drought, salinity and heat stresses. Transgenic plants overexpressing *GRPL1*-3HA showed significantly higher germination, root elongation and survival rate under salt stress. Moreover, the *35S::GRPL1-3HA* transgenic lines also showed higher survival rates under drought and heat stresses. *GRPL1* showed similar expression patterns with Abscisic acid (ABA)-pathway genes under different growth and stress conditions, suggesting a possibility that *GRPL1* might act in the ABA pathway that is further supported by the inability of ABA-deficient mutant (*aba2-1*) to induce *GRPL1* under drought stress. Taken together, our data presents *GRPL1* as a potential multi-stress responsive gene working downstream of ABA.

## Introduction

Being sessile in nature the plants are continuously exposed to different kinds of environmental stresses, including elevated level of salinity, drought and intense temperature [1] that causes reduction in crop productivity and pose a major threat to global food security [2]. The adverse effects of these abiotic stresses are further worsening by climate change, persistent reduction in the arable land and limiting water resources [3,4]. Extensive research in model plants and crops has aimed to understand the responses of plants to various biotic and abiotic stresses [5]. To meet the demands of growing population and varying climatic conditions of the globe there is a great urge to increase the global food production, consequently the demand of stress-tolerant crop varieties has been increased than in the past [6,7].

**Competing interests:** The authors have declared that no competing interests exist.

To cope with stress condition, plants have developed numerous adoptive strategies which include physiological, morphological, molecular and biochemical responses [8]. Plants have evolved several molecular mechanisms for adaptation and subsequent survival under stress conditions as revealed by stress induced transcriptomics. Hundreds of plant genes are differentially regulated in response to abiotic stresses, as demonstrated by RNA-seq analyses. A variety of stresses has been extensively studied in *Arabidopsis thaliana* [9]. RNA-seq has been proven as a powerful method for understanding the complex regulation networks and gene expression in many plant species responding to several kinds of biotic and abiotic stresses, such as Chinese cabbage [10], chickpea [11], maize [12], potato [13], *Ammopiptanthus mongolicus* [14] and soybean [15].

To mitigate the negative effects of these stresses, plants have evolved a sophisticated system of stress sensors, signaling transduction pathways and transcription factors (TFs) [16,17]. Over the past several years, the functional roles of glycine-rich proteins (GRPs) in the stress response have been investigated, and some members of this family in Arabidopsis have been shown to enhance seed germination and seedling tolerance to cold stress [18]. A large number of glycine rich proteins (GRPs) are also modulated by biotic and abiotic factors [19]. GRPs are also involved in responses to water stress, including drought and water logging. The grain yield of rice (*Oryza sativa*) during drought stress was improved following the incorporation of AtGRP2 and AtGRP7 into the rice genome [20].

In this study, we identified a novel gene *GRPL1* (*AT4G18280*) that is annotated as a glycine-rich cell wall protein-like protein in *Arabidopsis thaliana* by using publicly available transcriptomic data of plants exposed to different abiotic stresses. The co-regulation of *AT4G18280* with high-affinity K+ 5 (*HAK5*) under $K^+$ deficiency and salt stress has been reported [21]; however, no attempts were made to functionally characterize this gene. In this study, we characterized *GRPL1* by generating overexpression lines and its performance analyses under different abiotic stresses. Overpopulation of *GRPL1* results in increased tolerance to salt, drought and heat stresses in *Arabidopsis thaliana*. Manipulating the orthologues of this gene in economical important crops can be helpful to ensure the global food security under changing environmental conditions.

## Materials and methods

### RNA-seq public datasets retrieval and analysis using Tuxedo protocol

To gain insights of transcriptome wide genes expression, we selected publicly available transcriptome datasets of *Arabidopsis thaliana* plants based on the criteria that the samples used should be of similar developmental stage and ideally the level and duration of the specific abiotic stress should be similar. Based on this criterian, SRA009031 [22], GSE72806 [23], and SRP035234 [24] of *Arabidopsis thaliana* (Col-0) were downloaded and analyzed with the Tuxedo protocol [25]. Briefly, the raw reads were quality assessed, adopter trimmed and then aligned with TOPHAT2 followed by the transcript assembly with Cufflinks and differential gene expression analysis with Cuffdiff2. Downstream analysis and visualization of differentially expressed genes (DEGs) were done with custom R and Python scripts. Common target genes of different stresses were determined using Venny2 (available at: bioinfogp.cnb.csic.es/tools/venny).

### Plant material, growth conditions and stress response analyses

For all experiments Columbia accession (Col-0) of *Arabidopsis thaliana* was used. Two weeks-old plants grown under controlled conditions (16:8h day/night) were used for stress response experiments.

**Expression analysis.**   For transcript level expression of genes and validation of RNA-seq data, RNA was extracted from Arabidopsis seedlings using TRI Reagent (Sigma), and cDNA was synthesized from total RNA (2 μg) using Superscript-II reverse transcriptase (Invitrogen). Quantification was performed using LightCycler® 480 instrument. qPCR experiments were performed in three technical and biological replicates each. The average Ct values obtained from the qPCR reaction were used to calculate the relative quantity (RQ) for reference and gene of interest using the equation given below:

Relative Quantity (RQ) = $E^{(Ct[control]-Ct[treatment])}$

Where: E = Primer efficiency

T = Target gene

The relative quantities were then used to estimate the relative expression of the target gene using the equation:

Normalized Expression = $RQ_{sample(GOI)}/RQ_{sample(REF)}$

Where: GOI = Gene of interest

REF = Reference gene

Gene expression was normalized to Ubiquitin. Whereas, the protein gel blot was used for detection of HA-epitope-tagged proteins in the overexpression lines. For protein gel blot, anti-HA monoclonal antibody (Agrisera) was used. Ponceau staining was used as loading control.

**$PGRPL1$::$GUS$ cloning and GUS assay.**   To check the promoter activity of $GRPL1$ promoter, we cloned 1,758bp upstream region of $GRPL1$ translation start site using Infusion cloning system. Briefly, the $pBI101$ vector was linearized using $Sma$I restriction enzyme whereas the promoter region was amplified using a high fidelity pfu DNA polymerase using the primers mentioned in (S1 Table). The GUS assay was performed on the stable $T_3$ transgenic lines as described earlier [26]. Briefly, the samples were fixed with 90% acetone for 20min, washed and treated with staining buffer (containing X-Gluc) and incubated at 37˚C for 2h. After the removal of staining buffer and washing with ethanol the samples were observed under light microscope.

**Generation of over expression lines.**   The potential multi-stress tolerance $AT4G18280$ gene was amplified with gene specific primers (S1 Table) and cloned into a pUC19 background based binary vector containing a 35S Cauliflower Mosaic fused Virus (CaMV) promoter. Overexpression of GRPL1 was confirmed at transcript level via qPCR (S1 Table) and at protein level by western blot of the stable transgenic lines.

**Stress assays.**   To access the functionality of $AT4G18280$ as a potential multi stress tolerance candidate, performance of the overexpression lines of $T_4$ generation under salt, drought and heat stress was determined in comparison with wild-type plants. Salt stress was induced by addition of 100 and 200mM NaCl to the MS plates. Wild type and over expression lines were evaluated for fresh weight, germination percentage, relative root elongation, chlorophyll contents, proline content, melondialdehyde content (MDA) and survival rate (%). After completion of stress period, seedlings (10 plants/genotype) were taken from control and stressed pots and their fresh weights (mg/plant) were determined using a digital balance. For germination test, differences in the seeds germination under salt stress was recorded on the third day of transfer to growth chamber (22˚C). Relative root elongation (%) was measured as the percentage of root elongation under salt stress normalized to the control condition. Chlorophyll contents from leaves of the plants treated with 100 and 200 mM NaCl were quantified using a spectrophotometer (U-2810, Hitachi). After extraction of the supernatant from samples

homogenized in acetone the chlorophyll a and b were determined in totality in mg/g FW using the methodology described earlier [27]. Proline contents ($\mu$g/g FW) were quantified using the method reported earlier with minor modifications [28]. Melondialdehyde (MDA) content was measured by obtaining the supernatant from leaf samples of control and stressed plants through series of steps and then the obtained supernatant was used to measure the OD at 532, 600, and 450 nm and the MDA content was calculated using the method described by Zhang *et al.* [29].Survival rate (%) was determined by subjecting two week-old seedlings for NaCl (200 mM) treatment followed by calculating the alive plants percentage as described earlier [30].

Drought stress tolerance analysis was done by calculating the survival rate and water loss. Survival percentage was measures by subjecting 14-day old soil grown plants (WT and over-expression lines) to dehydration by withholding water supply for 20 days and then re-watered. Survival percentages were recorded 6 days after the recovery and experiment was repeated thrice [31]. Water loss analysis from detached leaves was performed as described earlier [32]. Briefly, ~0.5 g rosette leaves of three weeks old plants were weighed ($FW_0$) and incubated at 23˚C. The samples were then reweighed at 1, 2, 3, 4, and 5 hours ($FW_i$), whereas the dry weight (DW) was measured by drying the detached leaves at 80˚C for 5 hr. The water loss was calculated based on the initial fresh weight of plants using the formula as:

$[(FW_i-DW)/(FW_0-DW)] \times 100$ Where $FW_0$ and $FW_i$ are fresh weight for original and any given interval fresh weight, respectively, and DW is dry weight.

For heat stress tolerance analysis, the survival rate was measured after subjecting the plants to heat stress of 45˚C for 1 hr.

**Statistical analysis.**   Data of the physiological and biochemical parameters between the transgenic and non-transgenic lines were analyzed in triplicate through analysis of variance (ANOVA) using student t-test.

## Results

### Identification of *GRPL1* as a multi-abiotic stress-responsive gene

Adverse environmental conditions have been a major threat to global food security. The condition got worsens with the persistent reduction in the arable land, limiting water resources and climate change trends [33]. Therefore, we aimed to utilize the publicly available transcriptome data i.e. SRA009031 [22], SRP035234 [24] and GSE72806 [23] of plants exposed to different abiotic stresses in order to identify potential stress tolerance genes. Our analyses of these public transcriptome datasets identified a total of 760 genes that were commonly up regulated in response to heat, drought, salt and the combined heat and drought stress (Fig 1A and S2 Table). The top 10 multi-stress-induced genes of that list contained some heat shock proteins along with other known abiotic stress-responsive genes (Fig 1A). However, we selected *GRPL1* i.e. *AT4G18280* as a candidate because it showed strong induction in response to these abiotic stresses and was not studied earlier in the context of multi abiotic stress tolerance (Fig 1A). The expression level of *GRPL1* in response to different abiotic stresses is shown in Fig 1B. Though, all tested abiotic stresses showed induction of *GRPL1*, the PEG-induced osmotic and salt stress showed comparably higher levels of *GRPL1* mRNA than heat stress (Fig 1B). Consistent with the RNA-seq results, our qPCR analyses showed similar expression pattern for the given abiotic stresses (Fig 1C). Precisely, the mRNA levels of *GRPL1*gene was up-regulated by ~5.8 folds under salt stress, ~6.1 folds in the drought stress and ~4.1 folds in heat stress (Fig 1C). To further confirm this induction of *GRPL1* by abiotic stress-induced, we generated promoter *GRPL1*-driven GUS transgenic lines (*pGRPL1::GUS*). *pGRPL1::GUS* seedlings showed weak GUS staining under control conditions, indicating the weak promoter activity of *GRPL1*

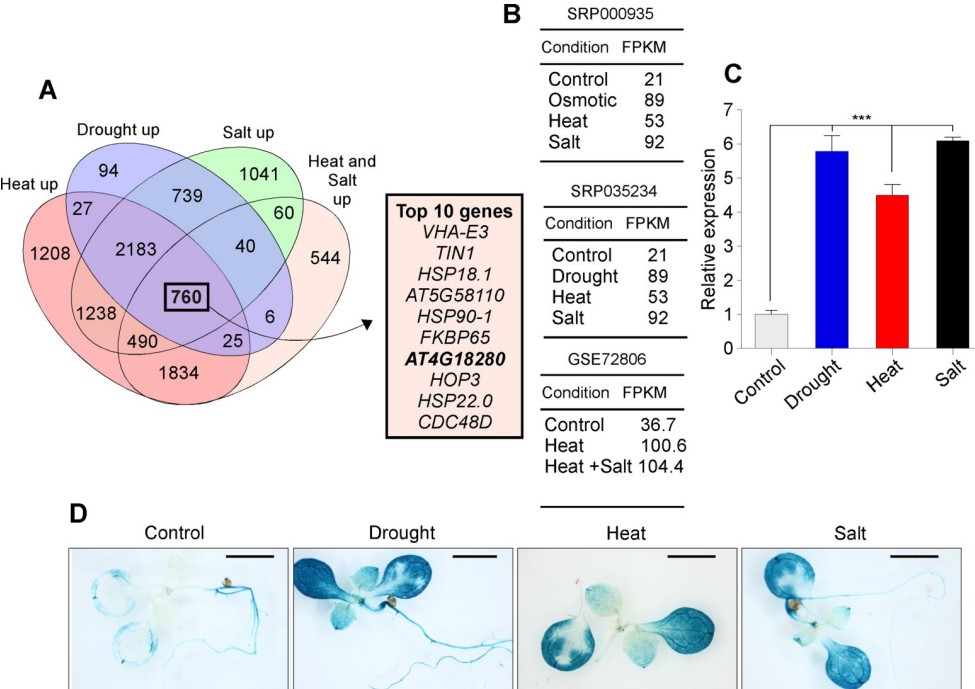

**Fig 1. The *GRPL1* expression is induced by abiotic stresses.** (A) Commonly up-regulated genes in response to multiple abiotic stresses. Table showing the top 10 upregulated genes among the total 760 genes. (B)Expression of *GRPL1* under drought, heat, salt and combined stress of heat and salinity, derived from public RNA-seq datasets. (C) qPCR validation of *GRPL1* induction under abiotic stresses (D) *pGPRL1::GUS* transgenic lines under control, drought, heat and salt stresses. Scale bar = 0.1 cm.

(Fig 1D), however, under abiotic stresses, we observed strong induction of *pGRPL1* promoter activity represented by the strong GUS staining. Overall, these results show that transcription of *GRPL1*is strongly induced under abiotic stresses, suggesting a potential role of *GRPL1*in modulating plant's response to abiotic stresses.

## Cloning and expression analysis of *AT4G18280*

To validate the potential role of *GRPL1* in abiotic stress response, we generated transgenic plants overexpressing HA-tagged *GRPL1*.First, the overexpression was analyzed at the transcript level by RT-PCR and at protein level by protein gel blot. Indeed, the three independent transgenic lines *1–4*, *4–3* and *6–2* showed higher mRNA (Fig 2A) and protein accumulation (Fig 2B). Also, the transgenic plants overexpressing *GRPL1* showed better growth and had significantly higher fresh weight compared to the WT plants (Fig 2C and 2D). These results confirmed the overexpression of *GPRL1* at both transcript and protein levels. The confirmed overexpression lines were tested under salt, drought and heat stress conditions to confirm their tolerance level.

## Function of *AT4G18280* overexpression under salt, drought and heat stress

To assess the effect of *GRPL1*overexpression, the transgenic lines were tested under salt, drought and heat stress conditions. As salinity is one of the major abiotic stresses that compromises crop productivity [34], we evaluated the performance of overexpression lines under 100 mM and 200 mM salt stress. Under control conditions, all lines showed 96–100% germination

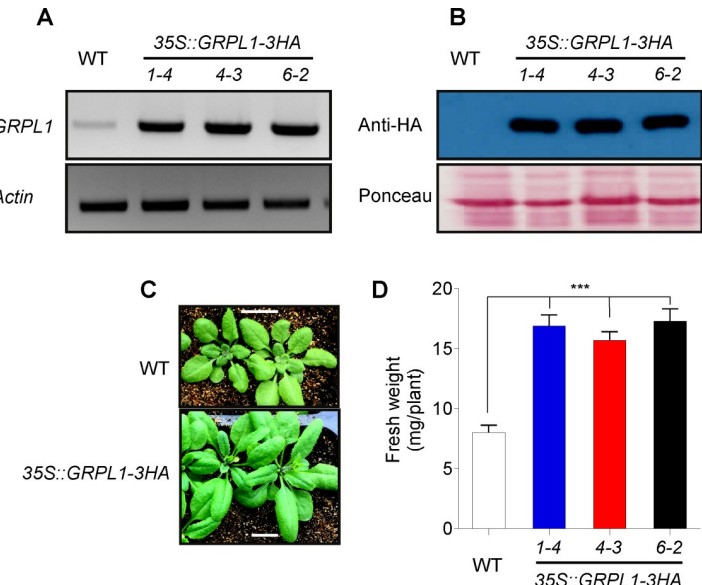

**Fig 2. Generation of *GRPL1* overexpression lines.** Confirmation of *GRPL1* overexpression through RT-PCR (A) and western blot (B). (C) Phenotype of *35S::GRPL1-3HA* transgenic lines. Scale bar = 1 cm. (D) Fresh weight comparison of WT and *35S::GRPL1-3HA* lines. 10 plants from each genotype were weighed.

(Fig 3A). However, the WT seeds grown on MS media supplemented with 100 mM and 200 mM NaCl showed reduced germination of ~55% and ~33%, respectively. Whereas, the overexpression lines showed better germination under both concentrations of salt stress i.e. about 75–85% of seeds successfully germinated under low (100 mM) while ~50–60% seeds germinated under high concentration (200 mM NaCl) salt stress (Fig 3A).

Consistent with the previous reports [35], we observed about 65% relative root elongation in WT plants whereas the overexpression lines showed significantly higher root elongation under stress condition i.e. up to 80% relative root elongation (Fig 3B). Salt stress induces production of reactive oxygen species (ROS), which damage the cellular components [36] including chlorophyll [37]. WT plants showed over 50% reduction in the overall chlorophyll contents under salt stress (Fig 3C). However, the *GRPL1* overexpressing plants showed significant tolerance towards salt stress as indicated by the reduced degradation of chlorophyll contents (~30–45%), highlighting the tolerance potential of the overexpression lines (Fig 3C). To further test the salt tolerance of *GRPL1* overexpression plants, we performed proline and MDA contents quantification. Proline acts as an osmolyte plays important roles in stabilizing macromolecules and membranes in cells by a higher accumulation [38]. The WT plants under salt stress accumulated ~2-fold higher proline contents whereas the overexpression lines showed ~4-fold higher proline accumulation (Fig 3D) implying that the transgenic plants are more tolerant to salt stress. However, performance of the transgenic line 1–4 was somewhat comparable with WT plants. The Melondialdehyde (MDA) level is an indicator of the cellular damage caused by stress conditions [39]. Although the *GRPL1*-mediated tolerance mechanism is still elusive, there is a possibility that overexpression of *GRPL1* may do so by the accumulation of proline and the efficient scavenging of ROS as represented by reduced quantities of MDA in transgenic plants (Fig 3D and 3E). Salinity is a serious threat to plants, disturbing all the physiological processes and even causes death of the plants [34]. Our results demonstrated that the overexpression of *GRPL1* enhanced the survival rate of plants under higher dose of salinity (Fig 3F). All of the overexpression lines showed significantly higher survival rates.

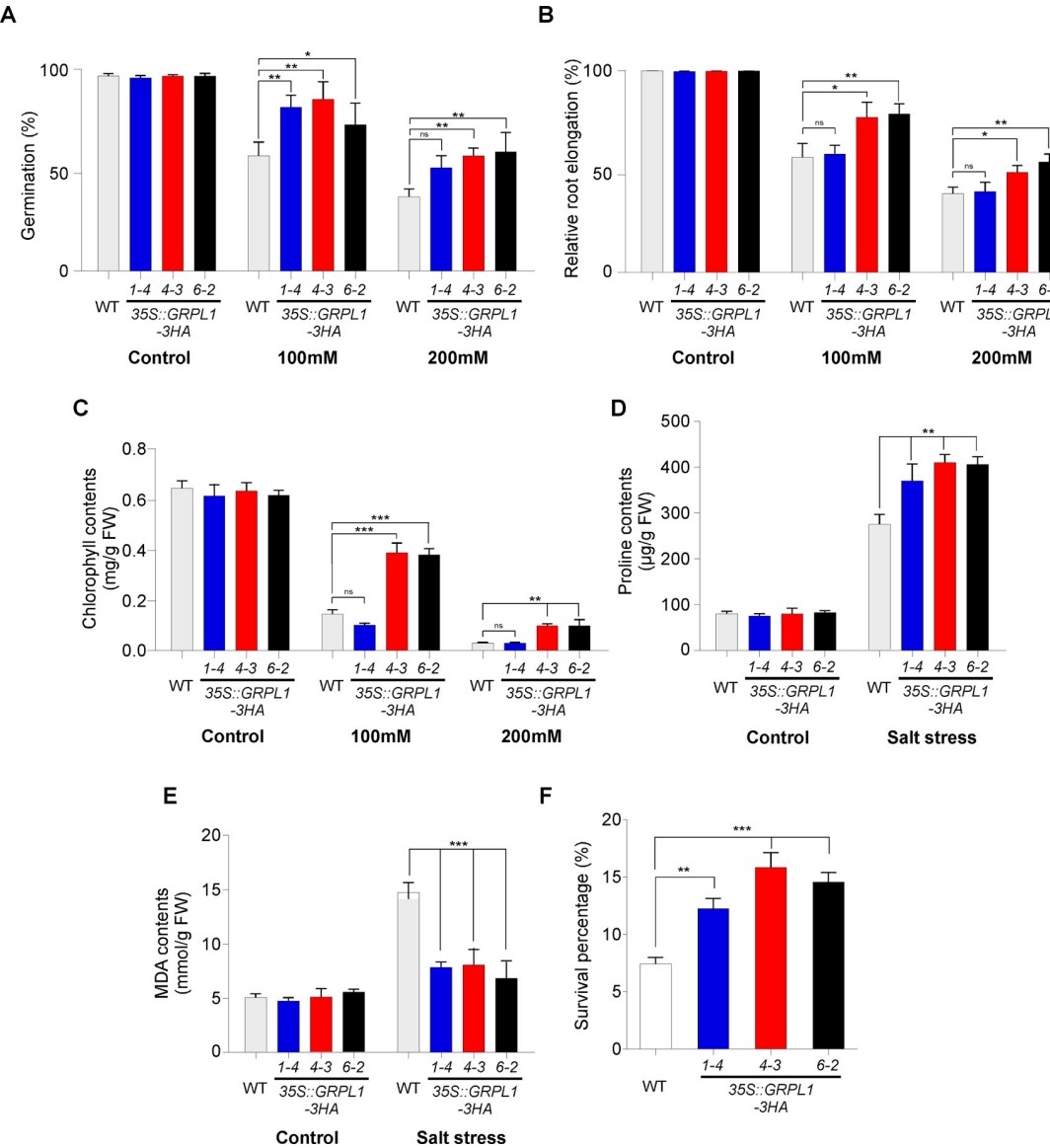

**Fig 3. Performance of *35S::GRPL1-3HA* transgenic lines under salt stress.** (A) Germination of three independent overexpression lines and WT seeds under control and under 100 and 200mM salt stressed conditions. (B) Relative root elongation and (C) Chlorophyll contents of *35S::GRPL1-3HA* lines in response to salt stress. (D) Higher accumulation of proline contents in *GRPL1* overexpression lines upon exposure to salinity stress. (E) Quantification of MDA contents and (F) Survival percentage of WT and transgenic plants exposed to salt stress.

Drought stress is also one of the biggest threats to food productivity. To examine the response of overexpression lines to drought stress, we subjected 14 days old seedlings to dehydration as no water was given to the plants for 20 days, followed by re-watering and checking the survival rates. Under water-deficient conditions for 20 days, only 22–27% of WT plants survived, whereas the overexpression lines showed significantly higher survival rates of 34–42% (Fig 4A and 4B). Furthermore, we also found the overexpression plants tend to retain more water contents compared to WT plants. After three hours of dehydration, the overexpression lines lost significantly lower amounts of water compared to WT plants and the difference between water loss got more significant with the increase in dehydration time (Fig 4C).

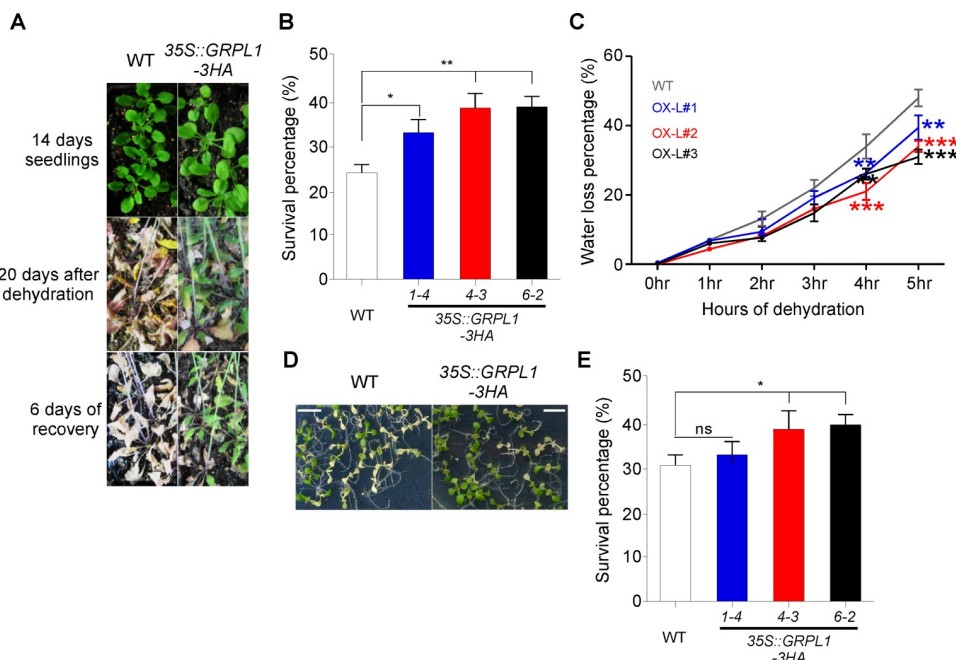

**Fig 4. Overexpression of *GRPL1* increases drought and heat-tolerance in transgenic plants.** (A) Phenotype of WT and transgenic lines in response to dehydration. (B) Bar graph representing the survival percentage of *GRPL1* overexpression lines in response in drought stress. (C) Line graph showing the water loss comparison between WT and transgenic lines upon exposure to dehydration. (D) Phenotype of WT and *GRPL1* overexpression lines under heat stress. Scale bar = 0.5 cm. (E) Survival percentage of WT and transgenic plants under heat stress.

These results suggest that overexpression of *GRPL1* results in reduced water loss under drought stress compared to WT plants and hence showed higher survival rates.

Re-analysis of public transcriptome data and qPCR analysis revealed induction of *GRPL1* under heat stress (Fig 1) which prompted us to check the performance of these overexpression lines under heat stress. Interestingly, the overexpression lines showed significantly higher survival rates compared to WT plants (Fig 4D and 4E) suggesting the possible role of GRPL1 in heat stress response too. However, the role of *GRPL1* in heat stress response requires further study.

Tolerance to abiotic stresses in plants is often mediated by ABA [39]. In order to check if *GRPL1* acts in the same pathway, we checked the expression of *GRPL1* in a collection of public microarray data. Interestingly, *GRPL1* showed similar expression patterns with the *ABA INSENSITIVE 1* (*ABI1*), *2*, and *5* (Fig 5A), suggesting a possibility that *GRPL1* might act in the ABA pathway. To further support this hypothesis, we used public transcriptome dataset (GSE75933) of ABA-deficient (*aba2-1*) mutant [40]. Reanalysis of RNA-seq data revealed *GRLP1* to be induced (~2.4-fold) by drought stress in WT background (Fig 5B), however, such stress treatment induction of *GRPL1* was compromised in the ABA-deficient mutants (*aba2-1*). Taken together, these results suggest that *GRPL1* may act downstream of ABA signaling.

## Discussion

Plants are continuously exposed to a wide range of biotic and abiotic stresses in their native environment. Extensive research in model plants and crops has aimed to understand the plant responses to a range of biotic and abiotic stresses, as these stresses reduce harvest yields [41]. In response to abiotic stresses, hundreds of plant genes are differentially regulated as

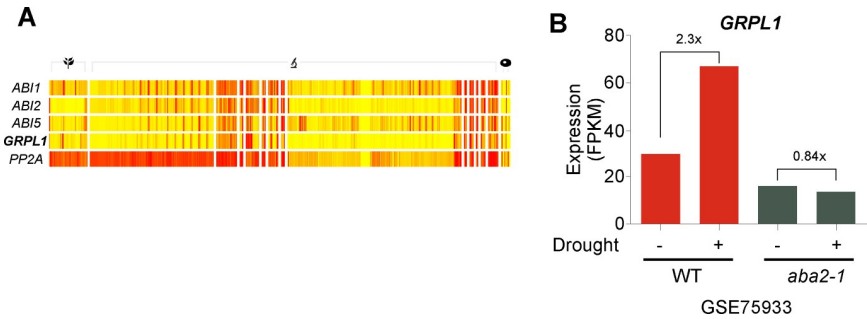

**Fig 5. Expression analysis.** (A) Profile of ABA pathway genes and *GRPL1* from a number of public microarray data sets (using ePLANT), *PP2A* was used as a control. (B) Expression of *GRPL1* in WT and *aba2-1* mutants under control and drought stress from the public RNA-seq data under the accession no GSE75933.

demonstrated by RNA-seq analyses. In this study, our basic understanding of the plant acclimation to abiotic stresses was extended through whole transcriptome (RNA-Seq) analysis based on the utilization of the publicly available transcriptome data sets. A total of 760 genes were commonly up regulated in response to heat, drought, salt and the combined heat and drought stress based on our analysis of the selected transcriptomic datasets. In order to identify the potentially multi stress responsive genes, we sorted the genes based on their fold-change values. The top 10 multi-stress-induced genes contained some heat shock proteins along with other known abiotic stress-responsive genes. However, we selected *GRPL1* i.e.*AT4G18280* as a candidate because it showed strong induction in response to these abiotic stresses and was not studied earlier in the context of multi abiotic stress tolerance. The RNA-seq predicted expression of the selected gene was confirmed by qRT-PCR, which showed that the results of both are significantly comparable to each other, and RNA-Seq results are reliable and can be used for further analysis. These results are consistent with the previous finding that the mRNA levels of glycine rich proteins (*GRPs*) are affected by different abiotic stresses, in a number of plant species [42].

The selected multi-stress induced gene was over-expressed in *Arabidopsis thaliana*, and its expression pattern was confirmed at transcript and protein level. The overexpression transgenic *Arabidopsis* lines with *AT4G18280* functional attributes were studied to confirm their role in different stress conditions. Significant differences in the *AT4G18280* overexpression lines and wild-type plants were observed under salt, drought and heat stresses. To confirm the tolerance level of over-expressed transgenic plants with *AT4G18280* gene under salt stress it was tested for some of the physiological parameters. The biological role of transgenic plants over-expressing the *AT4G18280* gene under salt stress was confirmed by measuring the fresh weight (mg/plant). Salt stress causes the reduction in fresh weights due to the relative increase in Na+ concentration by means of ionic stress. In our study, highest fresh weight was shown by all of the overexpression lines as compared to the wild type (Col-0) plants.

Germination is the most feasible approach used for selecting salt tolerance in plants [43]. A number of scientific studies demonstrated the negative effect of increasing dose of salinity on germination percentage of plants [44,45]. In our study, the overexpression lines evaluated for germination rate (%) under control and stress condition. The overexpression lines showed significantly high germination rate under salt stress condition than the wild type plants suggesting that the overexpression lines are able to germinate under salt stress and hence tolerant to reduce the detrimental effect on germination under increase dose of salt stress. This reduction in germination of WT seeds is consistent with the previous reports of salt stress negatively affecting germination rates, most probably by repressing biosynthesis of two phytohormones,

gibberellic acid (GA) and salicylic acid (SA) [46,47]. Overexpression of *GRPL1* exaggerated germination most probably through biosynthetic regulation of these two phytohormones. Being in direct contact with the soil, root length is considered one of the most important parameters to determine the salt stress tolerance of plants. It is well known that salt stress inhibits root meristem activity, cell cycle, and elongation of root cells in Arabidopsis and other plant species [48,49], resulting in retarded primary root growth [50,51]. It has been demonstrated that high dose of salt stress may reduce the root length by decelerating the water uptake by plants [52]. The relative root lengths determined in our study revealed that under stress condition (100 mM and 200 mM NaCl), the overexpression lines showed more root growth as compared to wild type plants which implies that the roots of overexpression lines are capable to grow even in high dose of salinity and hence are more tolerant. While similar pattern of root length was observed for overexpression lines as well as wild type plants under normal growth conditions. Soil salinity is a serious threat to plants, disturbing all the physiological processes and even causes death of the plants [34]. To know if the *AT4G18280* overexpression can rescue the low survival rate, survival rates were recorded for both the overexpression transgenic lines and wild-type plants after exposure to a salt stress of 200 mM NaCl. Our results showed that the overexpression lines were able to survive under higher dose of salinity as compared to wild type control plants and hence they are more tolerant to salinity.

Free proline content can increase upon exposure of plants to drought, salinity, cold, heavy metals or certain pathogens. Determination of free proline levels is a useful assay to monitor physiological status and to assess stress tolerance of higher plants [53]. Our results demonstrated the increase in proline content by all of the overexpression lines as compared to the wild type plants suggesting that they are more tolerant to salt stress. A large number of scientific studies have been conducted in *Arabidopsis thaliana* by overexpressing the native or antonym genes that resulted in higher proline content accumulation [54]. Different plants produce higher free proline levels in response to salinity and many scientists have cited the potential roles of proline such as stabilizing proteins, osmolyte, scavenging of hydroxyl radicals and regulating cytosolic pH [55–57]. It is known that salt stress causes membrane lipid peroxidation which is used as an indicator for salt stress induced oxidative membranes damage [58]. It is therefore used as an important criterion for plant stress tolerance evaluation under stress condition. In our study the melondialdehyde (MDA) content was determined both in control and salt stress conditions. Under salt stress condition all of the overexpression lines showed significantly lower MDA content as compared to the wild type plants. This imply that the transgenic plants overexpressing the *AT4G18280* gene confers tolerance to salt stress as they provide better protection against the oxidative damage. Plant tolerant to salt stress would accumulate less MDA contents compared to the susceptible plants [59]. Our results are consistent with the previous reports of reduced MDA accumulation in tolerant transgenic plants under salt and osmotic stresses [59]. Salt stress has been demonstrated to damage the chlorophyll contents of plants and hence reduces the photosynthesis of plants in different plant species [60,61]. In this study the overexpression lines and control plants were evaluated for chlorophyll content (mg/g FW) under control and salt stress condition. The overexpression lines maintain the chlorophyll content at high dose (200 mM NaCl) of salinity which means that they can withstand in saline environment by maintaining its chlorophyll content and hence more tolerant. Prolonged drought stress causes dehydration and a result death of the plants. To know the role of *AT4G18280* overexpression under drought stress, the overexpression lines with *AT4G18280* gene were tested for the survival rate. Significantly higher survival rate was recorded for all of the overexpression lines as compared to wild type control plants under drought stress. These results imply that the overexpression of Arabidopsis with *AT4G18280* gene results in greater tolerance to drought stress than the wild-type (Col-0) plants. Further, to support that *AT4G18280* overexpression is indeed involved

in providing drought tolerance to transgenic plants, they were tested for water loss at different intervals of dehydration. Our results showed that all of the overexpression lines were able to maintain its water content over increasing hours of dehydration. Significantly less water loss was recorded for the overexpression lines as compared to wild-type control plants; suggests that they are more tolerant to drought conditions.

After knowing the potential role of the over-expressed transgenic Arabidopsis plants with *AT4G18280* gene in salt and drought stress, it was evaluated to confirm its role in heat stress as well. As the heat stress is lethal to plants, causing a number of changes in plant's metabolism and even causes the mortality in plants. For this, the survival rate of the overexpression lines and wild type plants were recorded after subjected to heat stress of 45˚C for 1hr. The survival rate recorded for all the overexpression lines was significantly higher than the wild type plants which means that they are able to withstand heat stress and hence more tolerant.

The expression of *GRPL1* was checked in a collection of public microarray data and interestingly, *GRPL1* showed similar expression pattern with the *ABA INSENSITIVE 1* (*ABI1*), *2*, and *5*, suggesting a possibility that *GRPL1* might act in the ABA pathway. This hypothesis was further supported by analyzing public transcriptome dataset (GSE75933) of ABA-deficient (*aba2-1*) mutant [40]. *GRPL1* was highly induced under drought stress in WT background, however, such stress treatment induction of *GRPL1*was compromised in the ABA-deficient mutants (*aba2-1*). *ABA2* gene is an alcohol dehydrogenase and mutation in this gene results in blockage of xanthoxin to ABA aldehyde, a key step in ABA-biosynthesis resulting in low ABA production. ABA positively regulates the induction of drought-inducible genes. The signaling mechanism is well-known [62]. Briefly, ABA inhibits the PP2C-dependent negative regulation of the SnRK2 protein kinase [63], as a result the SnRK2 can phosphorylate its downstream target transcription factors (for instance ABFs) which in turn induces the target genes. The fact that the ABA-deficient mutant is unable to induce the expression of *GRPL1* in response to drought stress suggests that ABA is required for the induction of *GRPL1* and that it acts downstream of ABA. Taken together; these results suggest that *GRPL1* may act downstream of ABA signaling, however, further experiments are required to validate this hypothesis.

## Conclusions

In conclusion, using a number of publicly available transcriptomic datasets and transgenic approaches, we identified and characterized a previously unknown *GRPL1* as an abiotic stress-responsive gene. Constitutive expression of *GRPL1* in Arabidopsis increased plant's tolerance to salt, drought and heat stresses. Further understanding of the *GRPL1*function could be helpful in the devising breeding strategies for abiotic stress tolerance and can contribute to the global food security.

## Supporting information

**S1 Fig.**
(TIFF)

**S2 Fig.**
(TIFF)

**S1 Table. Primers used for RT-PCR.**
(DOCX)

**S2 Table. List of commonly up-regulated genes in the selected datasets.**
(DOCX)

## Acknowledgments

Authors are thankful to the contributions of each member of the Biochemical and Analytical Division in experiments, data analysis and manuscript preparation.

## Author Contributions

**Conceptualization:** Faiza Tawab, Iqbal Munir.

**Data curation:** Faiza Tawab.

**Formal analysis:** Zeeshan Nasim, Aqib Iqbal.

**Funding acquisition:** Iqbal Munir.

**Methodology:** Zeeshan Nasim.

**Project administration:** Iqbal Munir.

**Supervision:** Iqbal Munir, Mohammad Sayyar Khan.

**Visualization:** Adnan Nasim.

**Writing – original draft:** Faiza Tawab, Adnan Nasim, Mian Afaq Ahmad, Waqar Ali.

**Writing – review & editing:** Mohammad Sayyar Khan, Saleha Tawab, Raheel Munir, Maria Munir, Noreen Asim.

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
