## [Decision Letter · Decision Letter 0]

25 Sep 2020

PONE-D-20-26347

Identification and characterization of a novel multi-stress responsive gene in Arabidopsis

PLOS ONE

Dear Dr. Munir,

Thank you for submitting your manuscript to PLOS ONE. After careful consideration, we feel that it has merit but does not fully meet PLOS ONE’s publication criteria as it currently stands. Therefore, we invite you to submit a revised version of the manuscript that addresses the points raised during the review process.

We look forward to receiving your revised manuscript.

Kind regards,

Keqiang Wu, Ph.D

Academic Editor

PLOS ONE

Journal Requirements:

4. Please upload a new copy of Figure S1 as the detail is not clear. Please follow the link for more information: https://blogs.plos.org/plos/2019/06/looking-good-tips-for-creating-your-plos-figures-graphics/

Reviewers' comments:

Reviewer's Responses to Questions

**Comments to the Author**

1. Is the manuscript technically sound, and do the data support the conclusions?

Reviewer #1: Partly

Reviewer #2: No

2. Has the statistical analysis been performed appropriately and rigorously? 

Reviewer #1: Yes

Reviewer #2: Yes

3. Have the authors made all data underlying the findings in their manuscript fully available?

Reviewer #1: Yes

Reviewer #2: No

4. Is the manuscript presented in an intelligible fashion and written in standard English?

Reviewer #1: Yes

Reviewer #2: Yes

5. Review Comments to the Author

Reviewer #1: The authors carried out re-analysis of three publically available RNA seq datasets and zeroed in on At4g18280 (here named as GRPL1). The manuscript described the physiological analysis of GRPL1 over expression lines under salt, drought and heat stress. The study does add to knowledge on relatively less explored and functionally uncharacterized protein. However there are several points were this manuscript needs improvements.

My observation and comments are as follows.

• There is too much of redundancy in introduction section. There are many sentences with similar message or meaning, these needs to be removed. The aim of the study needs to be clearly mentioned.

• There is one study on identification and characterization of At4g18280 (here named as GRPL1) ‘Overexpression of a Novel Component Induces HAK5 and Enhances Growth in Arabidopsis’. Adams et al. 2013. The study suggested the role of this gene in pot deficiency and salt stress response. This ref is missing in the whole manuscript.

• It would be better to mention the plant /crop to which the SRA data sets belong. The reason or criteria for selecting only these three data sets out of many available for analysis should also be mentioned.

• Line 159-160. “leaves of three weeks old were excised to analyse the standardized water content”. What is the meaning of standardized water content??

• Line 187-192: As floral dip method is well known method, instead of detailed description just reference is sufficient to mention the transformation method used. Infact the very ref for floral dip is missing. The details of promoter cloning are also absent.

• There is repetition of methods in “Plant Material, Growth Conditions and Stress Treatment” and “Measurement of Physiological and Biochemical Indexes under Salt, Drought and Heat

Stress”. These can be combined under one heading.

• There is discrepancy in the number of days for drought stress imposition (water with holding) in Line no 156 (20 days), 231 (32 days) and 358 (28 days) .

• Result section:

• The authors described GO analysis of DEGs and next section started analyzing expression of AT4G18280 (GRPL1) which is too abrupt and requires explanation as to how the authors zeroed in on AT4G18280 (GRPL1). Moreover the first section “Global Gene …..Abiotic Stresses” details can be made shorter.

• Again there is redundancy in few lines of section 1 and section 2 of results.

• Line no. 277 PEG induces osmotic stress not drought stress. Moreover there is no distinction in the Table (Fig.1a) which is drought and PEG imposed stress.

• Line 295: First, the overexpression was analyzed at the…

• Fig. 2C should have scale bar for comparison of size. For Fig. 2D data from how many plants was recorded (replicates)? In general no of plants /replicates should be mentioned in all experiments

• Instead of “Over-expressed Transgenic Lines”, transgenic lines overexpressing GRPL1 or overexpression lines may be a better term.

• In Fig 3 B and C, over expression line 1-4 did not perform better than Wild type. so this should be mentioned instead of generalizing the observation. After how many days the survival % under salt stress was recorded? Should be mentioned in text.

• Line 350-353: without understanding of the biological function of GRPL1 , these are too presumptive statements.

• Lines 397 -406 these lines should be shifted to section 1 “Global Gene …..Abiotic Stresses” of results.

• Lines 410 -459: Discussion of DEGs is not required as these results might have been discussed by the authors of SRA study. Here authors should focus on how many genes they short listed infact a list of genes should have been more appropriate (in results). And then they can discuss validation of their selected gene GRPL1 by qRT-PCR (lines 459-462).

• Line 474-476: Authors have discussed propidium iodide staining for recording no of cells however there is no data presented. Similarly Line 528-529 electrolyte leakage observation under salt stress is mentioned however no data is given.

• Conclusion can be made more specific and short.

Reviewer #2: Reviewer 1

The author identified a gene GRPL1 which was induced by multiple stresses by querying transcriptome results of multiple stresses. The transgenic Arabidopsis was obtained by overexpression of GRPL1. It was proved that GRPL1 can improve the resistance to high salt, drought and heat stress through resistance identification and analysis of various physiological indexes. It is a new idea to identify genes involved in multiple stress resistance processes. However, the quality of the data in this paper is not high, especially the pictures are not standardized. It is difficult to understand that GRPL1 is regulated by ABA2 gene. Therefore, this paper can not be published in the PLOS ONE at present, and we hope to revise it. Suggestions for revision are as follows:

1. Why didn't we use GRPL1 mutants for phenotypic analysis.

2. When analyzing transcriptome data, the reason why GRPL1 gene was chosen is not clear.

3. Paragraph in P303-308 is too short, so it is suggested to combine with other results.

4. The discussion is too long.

5. There are conjectured results in the conclusion part, which are not supported by data, so it is suggested to remove them.

6. Fig2C is lack of scale, the size of pictures is different, and there is no clear picture of salt tolerance.

7. Fig.4A is not clear.

8. The primers of Tables 1 were incomplete and there were no QRT PCR primers.

9. ABA2 gene is alcohol dehydrogenase. It is difficult to understand why ABA2 mutation affects GRPL1 expression, because ABA2 is not a transcription factor.

6. PLOS authors have the option to publish the peer review history of their article (what does this mean?). If published, this will include your full peer review and any attached files.

Reviewer #1: No

Reviewer #2: No

---

## [Author Response · Author response to Decision Letter 0]

13 Nov 2020

Reviewer 1: 

1. There is too much of redundancy in introduction section. There are many sentences with similar message or meaning, these needs to be removed. The aim of the study needs to be clearly mentioned.

We thank Reviewer 1 for highlighting the redundancy in introduction section. We revised introduction section of the manuscript and also clearly included the aim of this study.

2. There is one study on identification and characterization of At4g18280 (here named as GRPL1) ‘Overexpression of a Novel Component Induces HAK5 and Enhances Growth in Arabidopsis’. Adams et al. 2013. The study suggested the role of this gene in pot deficiency and salt stress response. This ref is missing in the whole manuscript.

We thank Reviewer 1 for his suggestion, we cited the work of Adams et al., 2013 in the revised version.

3. It would be better to mention the plant /crop to which the SRA data sets belong. The reason or criteria for selecting only these three data sets out of many available for analysis should also be mentioned.

All the RNA-seq datasets belong to Arabidopsis thaliana, this information was added to the material and methods section (line83). A number of facts influenced the selection of these datasets, for instance, we wanted to have datasets that contain at least two or three biological replicates, the samples used should be of similar developmental stage, ideally the level and duration of the specific abiotic stress should be similar, and also our limited computational resources. We added this information in the methodology section (line 83-85).

4. Line 159-160. “leaves of three weeks old were excised to analyse the standardized water content”. What is the meaning of standardized water content??

We thank Reviewer 1 for pointing out this confusion. By standardized water content we meant “initial water content”. However, to make it clearer, we corrected and modified the water loss analysis methodology (line 152-158). 

5. Line 187-192: As floral dip method is well known method, instead of detailed description just reference is sufficient to mention the transformation method used. Infact the very ref for floral dip is missing. The details of promoter cloning are also absent.

We agree with Reviewer 1, as the floral dip is a commonly used and well-known procedure, we excluded the detailed description (line 187-191). We added details of promoter length and cloning strategy in the methodology (line 110-113).

6. There is repetition of methods in “Plant Material, Growth Conditions and Stress Treatment” and “Measurement of Physiological and Biochemical Indexes under Salt, Drought and Heat

Stress”. These can be combined under one heading.

As suggested by the Reviewer 1, we combined both “Plant Material, Growth Conditions and Stress Treatment” and “Measurement of Physiological and Biochemical Indexes under Salt, Drought and Heat Stress” sections under one heading of “Plant Material, Growth Conditions and Stress Response Analyses”.

7. There is discrepancy in the number of days for drought stress imposition (water with holding) in Line no 156 (20 days), 231 (32 days) and 358 (28 days) .

We thank Reviewer 1 for pointing out these typos, the correct number of days is 20. We did corrections in line no 231 (line 254 now and 156 (line 150 now). 

 Result section:

8. The authors described GO analysis of DEGs and next section started analyzing expression of AT4G18280 (GRPL1) which is too abrupt and requires explanation as to how the authors zeroed in on AT4G18280 (GRPL1). Moreover the first section “Global Gene …..Abiotic Stresses” details can be made shorter.

We thank Reviewer 1 for this valuable recommendation, we modified this section and also added more detailed description of selecting GRPL1 as a candidate gene (line 173-180).

9. Again there is redundancy in few lines of section 1 and section 2 of results.

As recommended by Reviewer 1, we combined section 1 and 2 of results 

10. Line no. 277 PEG induces osmotic stress not drought stress. Moreover there is no distinction in the Table (Fig.1a) which is drought and PEG imposed stress.

Correction was made to line no 277 (182 now) by changing “drought” to “osmotic”. Distinction was made to the table which now shows the drought and PEG imposed stress.

11. Line 295: First, the overexpression was analyzed at the…

We modified the sentence as suggested by Reviewer 1 (now line no 198)

12. Fig. 2C should have scale bar for comparison of size. For Fig. 2D data from how many plants was recorded (replicates)? In general no of plants /replicates should be mentioned in all experiments

We put the scale bar on each picture. For Fresh weight experiment, we weighed 10 plants for each genotype. We added this information in the figure legend and methodology section. 

13. Instead of “Over-expressed Transgenic Lines”, transgenic lines overexpressing GRPL1 or overexpression lines may be a better term.

As per the recommendation of Reviewer 1, “Over-expressed Transgenic Lines” was replaced with “overexpression lines” throughout the text. 

14. In Fig 3 B and C, over expression line 1-4 did not perform better than Wild type. so this should be mentioned instead of generalizing the observation. After how many days the survival % under salt stress was recorded? Should be mentioned in text.

AsReviewer 1recommended, we added the information of line 1-4 not performing better than WT plants (232-233). The survival percentage was recorded after two weeks of exposure to salt stress. We modified the text as well (line no 199). 

15. Line 350-353: without understanding of the biological function of GRPL1, these are too presumptive statements.

We agree with Reviewer 1 and toned-down our statement as below:

“Although the GRPL1-mediated tolerance mechanism is still elusive, there is a possibility that” (line 234-237)

16. Lines 397 -406 these lines should be shifted to section 1 “Global Gene …..Abiotic Stresses” of results.

We removed the redundant and out of the context part of discussion, to give more focus to significance of GRPL1 in stress tolerance 

17. Lines 410 -459: Discussion of DEGs is not required as these results might have been discussed by the authors of SRA study. Here authors should focus on how many genes they short listed infact a list of genes should have been more appropriate (in results). And then they can discuss validation of their selected gene GRPL1 by qRT-PCR (lines 459-462).

We thank Reviewer 1 for this suggestion and modified the discussion part accordingly.

18. Line 474-476: Authors have discussed propidium iodide staining for recording no of cells however there is no data presented. Similarly Line 528-529 electrolyte leakage observation under salt stress is mentioned however no data is given.

We thank Reviewer 1 for pointing out this mistake. We removed the description related to propidium iodide and electrolyte leakage under salt stress. 

19. Conclusion can be made more specific and short.

As suggested by Reviewer 1, we modified and shortened the discussion part (413-418)

Reviewer #2: 

The author identified a gene GRPL1 which was induced by multiple stresses by querying transcriptome results of multiple stresses. The transgenic Arabidopsis was obtained by overexpression of GRPL1. It was proved that GRPL1 can improve the resistance to high salt, drought and heat stress through resistance identification and analysis of various physiological indexes. It is a new idea to identify genes involved in multiple stress resistance processes. However, the quality of the data in this paper is not high, especially the pictures are not standardized. It is difficult to understand that GRPL1 is regulated by ABA2 gene. Therefore, this paper can not be published in the PLOS ONE at present, and we hope to revise it. Suggestions for revision are as follows:

1. Why didn't we use GRPL1 mutants for phenotypic analysis.

We did check two mutant lines (SALK_112442 and SALK_15781) for GRPL1, but unfortunately both mutant lines only showed a slight knock-down (~5-10%) of GRPL1. These lines have T-DNA insertion in 5′ and 3′ UTRs, respectively. Unfortunately, there are no available lines that has T-DNA insertion in the coding region. Therefore, we excluded them from further experiments. 

2. When analyzing transcriptome data, the reason why GRPL1 gene was chosen is not clear.

We thank Reviewer 2 for highlighting this point. We modified the section of results focusing on shortlisting of GRPL1 (line 173-180)

3. Paragraph in P303-308 is too short, so it is suggested to combine with other results.

We thank Reviewer 2 for this valuable suggestion. We combined this section of results with the previous heading.

4. The discussion is too long.

As recommended by the Reviewer 2, we modified the discussion part to make it more concise.

5. There are conjectured results in the conclusion part, which are not supported by data, so it is suggested to remove them.

We modified the conclusion part as recommended.

6. Fig2C is lack of scale, the size of pictures is different, and there is no clear picture of salt tolerance.

We thank Reviewer 2 for pointing this out, we added a scale bar on each picture of Figure 2C. We also updated the picture of salt tolerance (Fig. 4D) 

7. Fig.4A is not clear.

We understand the concern of Reviewer 2 but the objective of showing this picture was to show the difference in color. As from the figure one can see that the transgenic lines were still alive (greenish) compared to the wild-type plants (brownish).

8. The primers of Tables 1 were incomplete and there were no QRT PCR primers.

We thank Reviewer 2 for pointing out this mistake, we updated the list of primers in supplementary Table 1.

9. ABA2 gene is alcohol dehydrogenase. It is difficult to understand why ABA2 mutation affects GRPL1 expression, because ABA2 is not a transcription factor.

ABA2 gene is an alcohol dehydrogenase and mutation in this gene results in blockage of xanthoxin to ABA aldehyde, a key step in ABA-biosynthesis resulting in low ABA production. ABA positively regulates the induction of drought-inducible genes. The signaling mechanism is well-known. Briefly, ABA inhibits the PP2C-dependent negative regulation of the SnRK2 protein kinase (Umezawa et al., 2010, PCP), as a result the SnRK2 can phosphorylate its downstream target transcription factors (for instance ABFs) which in turn induces the target genes. 

The fact that the ABA-deficient mutant is unable to induce the expression of GRPL1 in response to drought stress suggests that ABA is required for the induction of GRPL1 and that it acts downstream of ABA. 

We added this information in Discussion section (line 402-409).

---

## [Decision Letter · Decision Letter 1]

23 Nov 2020

PONE-D-20-26347R1

Identification and characterization of a novel multi-stress responsive gene in Arabidopsis

PLOS ONE

Dear Dr. Munir,

Thank you for submitting your manuscript to PLOS ONE. After careful consideration, we feel that it has merit but does not fully meet PLOS ONE’s publication criteria as it currently stands. Therefore, we invite you to submit a revised version of the manuscript that addresses the points raised during the review process.

We look forward to receiving your revised manuscript.

Kind regards,

Keqiang Wu, Ph.D

Academic Editor

PLOS ONE

Reviewers' comments:

Reviewer's Responses to Questions

**Comments to the Author**

1. If the authors have adequately addressed your comments raised in a previous round of review and you feel that this manuscript is now acceptable for publication, you may indicate that here to bypass the “Comments to the Author” section, enter your conflict of interest statement in the “Confidential to Editor” section, and submit your "Accept" recommendation.

Reviewer #1: All comments have been addressed

Reviewer #2: All comments have been addressed

2. Is the manuscript technically sound, and do the data support the conclusions?

Reviewer #1: Yes

Reviewer #2: Yes

3. Has the statistical analysis been performed appropriately and rigorously? 

Reviewer #1: Yes

Reviewer #2: Yes

4. Have the authors made all data underlying the findings in their manuscript fully available?

Reviewer #1: Yes

Reviewer #2: Yes

5. Is the manuscript presented in an intelligible fashion and written in standard English?

Reviewer #1: Yes

Reviewer #2: Yes

6. Review Comments to the Author

Reviewer #1: Line 58 full form of GRPs should be mentioned (when it is mentioned first time)

Ilalics for gene names

Line 65-67 may be deleted

Method of calculation for qRT-PCR should be included in the expression analysis section

Mention vector background for overexpression construct, Generation at which the transgenic lines were analysed

Line 151 the word dessication should be replaced by dehydration or drought

FIG 2C the legend mentions scale bar as 1 cm however the picture does not appear to be in same scale, hence should be otherwise removed

Reviewer #2: (No Response)

7. PLOS authors have the option to publish the peer review history of their article (what does this mean?). If published, this will include your full peer review and any attached files.

Reviewer #1: No

Reviewer #2: No

---

## [Author Response · Author response to Decision Letter 1]

1 Dec 2020

Response to Reviewers

Reviewer #1

Line 58 full form of GRPs should be mentioned (when it is mentioned first time)

We thank Reviewer 1 for highlighting this; the full form of GRPs is now mentioned in line 58.

Ilalics for gene names

All gene names have been italicized.

Line 65-67 may be deleted

Lines 65-67 have been removed.

Method of calculation for qRT-PCR should be included in the expression analysis section.

The qPCR quantification is mentioned in lines 101-111.

Mention vector background for overexpression construct

The vector background is mentioned in line 129.

Generation at which the transgenic lines were analysed

It is mentioned in line 136.

Line 151 the word dessication should be replaced by dehydration or drought

The dessication word is replaced by dehydration in the manuscript.

FIG 2C the legend mentions scale bar as 1 cm however the picture does not appear to be in same scale, hence should be otherwise removed

Although the scale bar represents 1cm, their lengths are different in both images, i.e. the WT’s scale bar of 1cm has an actual size of 0.489cm whereas the one for transgenic plants is 0.351cm.

---

## [Editor Report · Decision Letter 2]

2 Dec 2020

Identification and characterization of a novel multi-stress responsive gene in Arabidopsis

PONE-D-20-26347R2

Dear Dr. Munir,

We’re pleased to inform you that your manuscript has been judged scientifically suitable for publication and will be formally accepted for publication once it meets all outstanding technical requirements.

Kind regards,

Keqiang Wu, Ph.D

Academic Editor

PLOS ONE
---

## [Editor Report · Acceptance letter]

9 Dec 2020

PONE-D-20-26347R2 

Identification and characterization of a novel multi-stress responsive gene in Arabidopsis 

Dear Dr. Munir:

I'm pleased to inform you that your manuscript has been deemed suitable for publication in PLOS ONE. Congratulations! Your manuscript is now with our production department. 

Kind regards, 

on behalf of

Professor Keqiang Wu 

Academic Editor

PLOS ONE